# College from home during COVID-19: A mixed-methods study of heterogeneous experiences

**Margaret E. Morris**[1]*, **Kevin S. Kuehn**[2], **Jennifer Brown**[3], **Paula S. Nurius**[4], **Han Zhang**[5], **Yasaman S. Sefidgar**[5], **Xuhai Xu**[1], **Eve A. Riskin**[6], **Anind K. Dey**[1], **Sunny Consolvo**[7], **Jennifer C. Mankoff**[5]

1 Information School, University of Washington, Seattle, Washington, United States of America,
2 Department of Psychology, University of Washington, Seattle, Washington, United States of America,
3 School of Public Health, University of Washington, Seattle, Washington, United States of America,
4 School of Social Work, University of Washington, Seattle, Washington, United States of America,
5 Computer Science & Engineering, University of Washington, Seattle, Washington, United States of America, 6 Department of Electrical & Computer Engineering, University of Washington, Seattle, Washington, United States of America, 7 Google, Mountain View, California, United States of America

* margiemm@uw.edu

**Data Availability Statement:** All quantitative data and analysis scripts are available on GitHub (https://github.com/kskuehn/UWEXP_COVID). The manuscript includes extensive quotes from

## Abstract

This mixed-method study examined the experiences of college students during the COVID-19 pandemic through surveys, experience sampling data collected over two academic quarters (Spring 2019 $n_1$ = 253; Spring 2020 $n_2$ = 147), and semi-structured interviews with 27 undergraduate students. There were no marked changes in mean levels of depressive symptoms, anxiety, stress, or loneliness between 2019 and 2020, or over the course of the Spring 2020 term. Students in both the 2019 and 2020 cohort who indicated psychosocial vulnerability at the initial assessment showed worse psychosocial functioning throughout the entire Spring term relative to other students. However, rates of distress increased faster in 2020 than in 2019 for these individuals. Across individuals, homogeneity of variance tests and multi-level models revealed significant heterogeneity, suggesting the need to examine not just means but the variations in individuals' experiences. Thematic analysis of interviews characterizes these varied experiences, describing the contexts for students' challenges and strategies. This analysis highlights the interweaving of psychosocial and academic distress: Challenges such as isolation from peers, lack of interactivity with instructors, and difficulty adjusting to family needs had both an emotional and academic toll. Strategies for adjusting to this new context included initiating remote study and hangout sessions with peers, as well as self-learning. In these and other strategies, students used technologies in different ways and for different purposes than they had previously. Supporting qualitative insight about adaptive responses were quantitative findings that students who used more problem-focused forms of coping reported fewer mental health symptoms over the course of the pandemic, even though they perceived their stress as more severe. These findings underline the need for interventions oriented towards problem-focused coping and suggest opportunities for peer role modeling.

participant interviews to substantiate the thematic analysis. Out of caution for participant privacy and security, the qualitative data cannot be made publicly available. Data access queries may be directed to Kelli Jo Fullan, Assistant to the Dean, Information School, University of Washington (contact via kjfullan@uw.edu).

**Funding:** This material is based upon work supported by the National Science Foundation under Grant Nos. EDA-2009977 awarded to JM, PN, AD, and ER, CHS-2016365, and CHS-1941537, awarded to JM. Funding was also provided through a grant from Samsung awarded to JM, PN, AD, and ER. KSK (the second author) was supported by a National Research Service Award from the National Institute of Mental Health (F31MH117827). Additional support came from a Google Security and Privacy unrestricted gift awarded to JM that provided consulting income for researcher MM. SC, a researcher at Google, is an author on this paper who contributed to study design, analysis, and manuscript preparation. The specific role of this author is articulated in the 'author contributions' section. The manuscript was approved by an internal Google review.

**Competing interests:** The support received from Google, the involvement of a researcher employed by Google, and the grant from Samsung do not alter our adherence to PLOS ONE policies on sharing data and materials.

## Introduction

Remote education became the default for most undergraduates in the U.S. in the Spring of 2020 due to the COVID-19 pandemic and may continue to be an important facet of university education in the future. At the time of this publication, many colleges and universities are announcing plans for instruction in Spring of 2021 to remain mostly online. Further, remote education may be preferred by students with disabilities, by international students, by those who are taking classes while working, and by those who cannot easily move out of family homes due to financial pressures or caretaking responsibilities. Additional pandemics in the near future, predicted by public health experts, also increase the probability that remote college education will continue and expand in upcoming years [1, 2].

As a consequence of the pandemic, many U.S. college students—particularly undergraduates—returned home to live with parents or other caregivers in the Spring of 2020. There may be developmental costs of this change because college, and specifically the move from parents' homes to campus, is often considered a transition towards independence and adulthood. Returning to living with family during this period has been anticipated to interfere with independence in expression and participation in social, religious, and political activities [3]. Learning and social belongingness may also suffer. Interactions with faculty and peers, both within and outside of classrooms, contribute to learning and engagement [4]. Participation in campus organizations and activities, which similarly extend learning and belonginess, may be particularly valuable for minority students [5].

Research on psychosocial distress during the pandemic is polarized between indications of major mental health sequelae and signs of resilience. Comparing June 2020 survey responses to 2019 surveys, Czeisler et al. [6] found increased symptoms of depression, anxiety and traumatic distress. Czeisler et al. [6] also find a notable increase in suicidal ideation among 18–24 year-olds during the COVID-19 pandemic, although it is worth noting other researchers have not found increases in suicidal thoughts and behaviors [7, 8]. Based on weekly surveys of 217 college students enrolled in the Student Life study, Huckins et al. [9] report increased depression and anxiety symptoms in the Winter 2020 term compared to terms in previous years, and an increase in these symptoms over the Winter term. Depression and anxiety were particularly common during the pandemic among students who reported difficulty adapting to remote education, based on a survey of 30,725 undergraduate students conducted in May-July 2020 at nine public research universities [10]. Deteriorated mental health from the pandemic was anticipated given its widespread and long-term interference with social systems, income and other resources [3]. Contrasting indications of resilience and adaptability come from Kuczynski et al. [11] who found no rises in self-reported anxiety, depression or loneliness during the pandemic, based on daily surveys of approximately 500 adults. Folk et al. [12] similarly found little changes in feelings of social connectedness among 467 college students surveyed in both February and April of 2020, and a slight drop in loneliness scores among 386 community adults surveyed at these two time points. These studies suggest either distress or resilience in response to the pandemic; they do not describe individual variation in distress, or examine how distress relates to specific struggles or coping strategies.

Negative impacts of the pandemic on college students may depend on socioeconomic vulnerabilities. Disparities in technology access and related technical and social infrastructure may have an increased impact when all learning is online. Setbacks due to the COVID-19 pandemic add to factors working against low socioeconomic status (SES) students and may impede educational trajectories [13]. Evidence of this negative impact on disadvantaged and underrepresented students is already emerging. Students from lower-income backgrounds are 55% more likely to have delayed graduation due to the pandemic and 41% more likely to

report that the pandemic impacted their choice of major [14]. Distractions and family obligations were reported by minority students before the pandemic [15] and are also disproportionately burdening minority students during the pandemic [16].

Adverse effects from the pandemic may also result from the strategies students employ to regulate stress. Disengagement forms of coping, often characterized as avoidance, distraction, and suppression, reduce negative emotions brought on by stressful events, but do not address the underlying causes of stress. For example, procrastinating on a difficult assignment pushes away anxiety, but only for the short term. Disengagement coping is linked to higher rates of psychopathology in adolescents [17]. Early evidence from a cross-sectional analysis suggests that problem-focused strategies (i.e., active coping techniques such as seeking advice or cognitive reappraisal) may protect against the deleterious effects of COVID-related stress [18].

To more fully understand college students' experiences during the pandemic, we conducted a mixed-methods study including thematic analysis of interviews and quantitative analysis of survey data. Interviews and surveys inquired about challenges that students experienced and their strategies for adjusting. Approximately 25% of these students were of low socioeconomic status (SES) and therefore potentially at greater risk of disrupted learning than other students. This research, conducted at one of the first U.S. universities to cancel in-person classes, provides a window into the experiences of college students coping with the pandemic.

This study addressed the following research questions: (1) Did psychosocial wellbeing of college students worsen during the pandemic, specifically during remote education? (2) Did anxiety, depressed affect and loneliness increase over time, as students spent more time online with fewer in-person interactions? (3) Were students' experiences uniform? (4) Were some students more prone to psychological and academic distress? (5) What challenges did students experience in the context of remote learning during the pandemic and how did they cope with these challenges?

## Methods

### Participants

This analysis drew from a multi-year study focused on stress and well-being among college students [19]. For the past three years, undergraduate students were recruited to participate in a study for the duration of at least one academic term at a large public university in the Pacific Northwest (see Table 1 for term dates). Data from 2019 and 2020 are included in this analysis. Data collection for both 2019 and 2020 started during the break immediately before Spring term; in 2020 this was after the start of the COVID-19 pandemic in the U.S. (see timeline in Table 1). Some participants continued on with the project through multiple years; however, new participants were recruited each year.

**Table 1. Timeline of local response to the pandemic.**

| Date | Response |
|---|---|
| 2/28/2020 | Alerts regarding local community spread of COVID-19. |
| 3/6/2020 | University announced shift to online learning for the remainder of Winter term (6 January 2020–20 March 2020). |
| 3/13/2020 | University announced that Spring term (30 March 2020–12 June 2020) would begin online. Statewide ban on large gatherings and closure of K-12 schools. |
| 3/18/2020 | University announced that all of Spring term would be online. |
| 3/23/2020 | Statewide stay-at-home order issued. |
| 5/4/2020—present (March 2021) | Phased restrictions on in-person contact, based on county public health metrics. Classes remained online for the entire 2020–2021 academic year. |

See S1 Table for participant demographics. Briefly, the 2019 sample included 253 (31.2% white; 49.8% women) participants. The 2020 sample included 147 (28.6% white; 55.1% women) participants. The majority of 2020 participants (71.6%) participated in one or two previous years of the study; 90 completed Time 1 (intake survey) and experience sampling method (ESM) surveys in both the 2019 and 2020 studies. Recruitment, focused on first-year engineering and computer science students, used online discussion forums, social media, and departmental email lists.

Interview participants, recruited from the 2020 cohort described above, were invited to participate based on two primary selection criteria: 1) *low SES*, using a modified version of the MacArthur Scale of Subjective Social Status [20] and 2) *ethnicity* (African American, Asian, Latinx, Native American/Alaska Native, or Pacific Islander). Although not required for participation, most participants were living with their families at the time of interviews. Exceptions include two participants who moved to their families' homes early in the pandemic but later moved back to the vicinity of campus to live with other students. Three participants had already been commuting to campus from their families' homes before the pandemic began.

## Procedures

We conducted a mixed-methods study that included quantitative survey and experience sampling data in addition to semi-structured qualitative interviews. The procedures of both surveys and interviews are described below.

**Survey and experience sampling procedures.**  Participants completed a self-report, online intake (Time 1) and exit (Time 2) survey. Both Time 1 and Time 2 surveys were administered via Qualtrics. Participants were emailed a link and instructed to complete the Time 1 survey during the break before Spring term. Participants were emailed a link for the Time 2 survey at the end of Spring term and instructed to complete it after their last final exam. Each of the Time 1 and 2 surveys took approximately one hour to complete, and participants were asked to finish all questions in one sitting. For ESM, participants received two surveys per week delivered to their cell phones on Wednesdays and Sundays. Details are provided in S2 Table. Written consent was obtained from participants in this research, which was approved by the institutional review board (Number 00003244).

**Interview procedures.**  Semi-structured individual interviews addressed experiences with remote education and psychosocial wellbeing during the pandemic. Interviews were conducted from June 8 to June 24, 2020. We inquired about participants' experiences with remote education including online classes, group exercises, study groups, and office hours, and technological barriers to learning. We also discussed their adjustment to studying and living in the family home. We asked participants to describe changes in their communication as they avoided in-person socializing (i.e., "physical distancing"). We asked about how they communicated with close friends and family, less close relations, and communities.

Interviews, approximately 90 minutes in length each, were conducted remotely over Zoom video conferencing and were audio recorded and transcribed for analysis. Participants' names were not used during interviews, nor were they associated with transcripts or audio recordings. The interviews were conducted by one or two researchers. Researchers and participants had video on during the interview for rapport, but only audio files were saved. Participants received $40USD for their participation. Oral consent was obtained from participants before beginning interviews. This research was approved by the institutional review board (Number 00009798).

## Measures

**Time 1 and Time 2 surveys.**   *Depressive symptoms*. Depressive symptoms were measured with the Beck Depression Inventory—II [21] (BDI-II; α's = .90-.91). *Anxiety*: Anxiety indicators were measured with the trait subscale of the State-Trait Anxiety Inventory [22] (STAI; α's = .90-.91). *Loneliness*: Loneliness was measured with the UCLA Loneliness Scale [23] (short-form; α's = .86-.90). *Stress*: Stress was measured with the Perceived Stress Scale [24] (PSS; α's = .82-.85).

**Experience sampling measures.**   *Negative affect*. Participants were asked "How are you feeling right now?" regarding their feelings of depression, anxiety, frustration, and fear. Participants rated their emotions on a 5-point scale ranging from 0 (*"None"*) to 4 (*"Extremely"*). Items were summed across these four affective states to create a measure of composite negative affect. This composite measure of negative affect was very reliable between participants and across ESM prompts in this sample ($R_{kf}$ = .97).

*Loneliness*. Participants rated their current feelings of loneliness on a single item with Likert scale ranging from 0 (*"None"*) to 4 (*"Extremely"*). The item demonstrated large variability via intra-class correlation across observations (ICC = .37).

*Depressive symptoms*. Depressive symptoms were assessed once per week with the 4-item PHQ-4 [25] ($R_{kf}$ = .98). *Stress*: Stress was assessed once per week with the 4-item PSS-4 [24] ($R_{kf}$ = .91).

*Stressful experiences*. A checklist assessed a variety of stressful experiences that participants encountered on the day of the survey. Specific forms of stress (such as academic concerns, financial stress, barriers to learning, and caregiving responsibilities) were separated and dichotomized such that "1" indicated that a participant had experienced that form of stress and "0" indicated a participant had not experienced that form of stress. These items varied substantially across reports (ICCs = .28 -.42). Participants were asked the severity of the most stressful experience on a 4-point Likert scale ranging from 1 (*"Not at all stressful"*) to 4 (*"Very stressful"*).

*Coping behaviors*. Coping in response to stress was assessed via a single-item worded, "What was your strategy for managing the stress just described?". Participants were instructed to select all the strategies they used from a checklist of 19 different strategies (ICCs = .29 -.42). For purposes of analyses, we created a variable that summed across all the problem-focused strategies (e.g., "Created a plan," "Focused on the positive aspects," etc.) and reverse coded disengagement strategies (e.g., "Avoided/ignored the situation," "Gave up," etc.). Participants were also asked how successful the various strategies they used in response to stress were on a 5-point Likert scale ranging from 1 (*"Not at all"*) to 5 (*"Extremely"*).

## Analytic approach

**Analysis of survey and Experience Sampling Method (ESM) data.**   To examine pandemic-related and within-year changes, we compared mean levels in Time 1 and Time 2 surveys from the prior year to the current cohort (i.e., March 2019 to March 2020 and June 2019 to June 2020) and compared within-person changes from the 90 participants who were in both the 2019 and 2020 cohorts. We then compared mean levels in Time 1 and Time 2 surveys of 2020 data in linear regression models. Next, we examined mean levels in 2019 and 2020 ESM data to test change over time using hierarchical linear models [26]. Based on prior reports of increased mental health concerns during the COVID-19 pandemic [6, 9, 10], we examined whether students who reported greater distress at Time 1 continued to experience more adversity throughout the term. Finally, participants' coping responses over the course of the 2020 study were examined. To disaggregate within- and between-person effects, the coping

strategies variable was centered, using grand mean centering as well as within-person centering [27]. A between-person effect would suggest that individuals who relied more on problem-focused coping differed from those who relied on disengagement coping in their experiences of distress. A within-person effect would indicate that *when* someone used more problem-focused strategies, relative to their own average, they experienced significantly more or less distress. Due to the large number of effects we examined (185), we used the Benjamini-Hochberg procedure [28] to limit the possibility of false positives. For this procedure we set the false discovery rate at 5%.

*Missing data*. We used listwise deletion to handle missing data for all analyses due to the high rate of ESM compliance (compliance rate = 95.34%). Listwise deletion is not known to bias parameter estimates when there is a low rate of missingness (< 10%) [29].

**Analysis of interviews.** Interviews were analyzed according to principles of thematic analysis [30]. One researcher inductively identified a preliminary set of codes and related themes after conducting and summarizing interviews. An interdisciplinary team then refined and expanded the preliminary set of codes in a close review of transcripts and interview summaries. All transcripts were coded by at least two researchers. After sets of 3–5 transcripts were reviewed by two or three coders, the team discussed and agreed on refinements such as adding, removing and combining codes. The goal of this iterative process was to build consensus on the codes and their application. Disagreements were resolved by clarifying the code criteria. In cases when two researchers used different but related codes, all codes were applied to that line of the transcript. Examining code expression across participants allowed researchers to derive the higher-level themes shared in results.

## Quantitative results

### Between- and within-person comparisons from 2019 to 2020

Comparing Time 1 data from 2019 and 2020, participants in 2020 did not report different Time 1 levels of depressive symptoms ($F$ [df = 1, 398] = 0.05, $p$ = .82), anxiety ($F$ [df = 1, 398] = 1.89, $p$ = .17) stress ($F$ [df = 1, 398] = 0.91, $p$ = .34) or loneliness ($F$ [df = 1, 398] = 0.72, $p$ = .40) (See S3 Table). Time 2 comparisons from 2019 and 2020 similarly indicate no yearly differences in depressive symptoms, anxiety, stress, or loneliness. Comparing the subset of individuals in both 2019 and 2020 data, there were within-person changes in anxiety ($F$ [df = 1, 89] = 6.29, $p$ = .01), however, anxiety indicators in 2020 were lower than in 2019 at Time 1. There were no within-person changes in Time 1 depressive symptoms, stress, or loneliness nor were there any within-person differences in any of these variables, including anxiety, in Time 2 data.

### Within-person comparisons over Spring 2020 term

Comparing Time 1 to Time 2 data in 2020 (S4 Table), participants did not report significant linear within-person changes over the course of the term in measures of depressive symptoms ($F$ [df = 1, 141] = 3.81, $p$ = .053), anxiety ($F$ [df = 1, 141] = 2.79, $p$ = .10), stress ($F$ [df = 1, 141] = 0.15, $p$ = .70), or loneliness ($F$ [df = 1, 141] = 0.36, $p$ = .55). 2020 ESM reports of distress over time, however, did slightly increase (S5 Table). Specifically, mean levels of depressive symptoms (b = 0.07; $t$ [df = 2259] = 4.57, $p$ < .001), stress (b = 0.04; $t$ [df = 2259] = 2.80, $p$ < .01), depressed affect (b = 0.02; $t$ [df = 2259] = 4.86, $p$ < .001) anxious affect (b = 0.01; $t$ [df = 2259] = 2.40, $p$ < .01), and composite negative affect (b = 0.02; $t$ [df = 2259] = 5.83, $p$ < .001) all increased over the duration of the quarter. Although statistically significant, the rate of change was very small (b's = 0.01 to 0.07) suggesting increases of up to one percent with each observation.

## Within-person comparisons over Spring 2019 and 2020 terms

The lack of group differences in mean levels of Time 1 and Time 2 responses between 2019 and 2020 led to the question of whether some students were more negatively impacted by the pandemic and remote education than others. We tested models using Time 1 reports of mental health indicators as predictors of distress using ESM data of the 90 participants who were in both the 2019 and 2020 cohorts (S6–S8 Tables). For clarification, the following analyses use Time 1 survey responses to examine changes in ESM data over the course of the 2019 and 2020 phases. In pooled analyses, Time 1 levels of depressive symptoms predicted increased depressive symptoms ($\beta = 0.40$, $t$ [df = 1299 = 10.97, $p < .001$); stress ($\beta = 0.29$, $t$ [df = 1262] = 7.89, $p < .001$); depressed affect ($\beta = 0.32$, $t$ [df = 2655] = 10.31, $p < .001$); anxious affect ($\beta = 0.17$, $t$ [df = 2655] = 5.41, $p < .001$); loneliness ($\beta = 0.26$, $t$ [df = 2655] = 7.75, $p < .001$); and composite negative affect over time ($\beta = 0.34$, $t$ [df = 2655] = 10.76, $p < .001$). This implies that a standard deviation increase in depressive symptoms at Time 1 was associated with a .17 -.40 standard deviation increase in distress over the course of the study. Time 1 levels of anxiety also predicted increased depressive symptoms ($\beta = 0.43$, $t$ [df = 1299] = 11.80, $p < .001$); stress ($\beta = 0.41$, $t$ [df = 1262] = 11.12, $p < .001$); depressed affect ($\beta = 0.37$, $t$ [df = 2655] = 11.34, $p < .001$); anxious affect ($\beta = 0.19$, $t$ [df = 2655] = 5.74, $p < .001$); loneliness ($\beta = 0.26$, $t$ [df = 2655] = 7.58, $p < .001$); and composite negative affect ($\beta = 0.50$, $t$ [df = 2655] = 15.20, $p < .001$). Finally, Time 1 levels of loneliness predicted increased depressive symptoms ($\beta = 0.25$, $t$ [df = 1299] = 7.15, $p < .001$); stress ($\beta = 0.19$, $t$ [df = 1262] = 5.44, $p < .001$); depressed affect ($\beta = 0.14$, $t$ [df = 2655] = 4.78, $p < .001$); loneliness ($\beta = 0.15$, $t$ [df = 2655] = 4.86, $p < .001$); and composite negative affect ($\beta = 0.23$, $t$ [df = 2655] = 7.62, $p < .001$). Put simply, these models imply that year-over-year, individuals reporting higher indicators of depression, anxiety, and loneliness at the initial assessment continued to experience higher levels of distress throughout the term.

To examine pandemic-related effects, we tested an exploratory three-way interaction between the year of the study (i.e., 2019 versus 2020), day of the study, and Time 1 levels of mental health indicators (S9–S11 Tables and Fig 1). This allowed us to test whether distress increased at a faster rate in 2020 than in 2019, for those participants who were higher in depressive symptoms, anxiety, or loneliness at Time 1. For Time 1 levels of depressive symptoms, the three-way interaction was significant in predicting depressed affect ($\beta = 0.04$, $t$ [df = 2649] = 5.40, $p < .001$) and loneliness ($\beta = 0.02$, $t$ [df = 2649] = 2.69, $p < .01$). That is, depressed affect and loneliness increased more quickly in 2020 than in 2019 for those who reported higher depressive symptoms at Time 1. For the three-way interaction between Time 1 anxiety, day, and year of the study, there was only a significant interaction in predicting depressed affect ($\beta = 0.03$, $t$ [df = 2649] = 4.03, $p < .001$). In other words, depressed affect increased more quickly in 2020 than in 2019 for those who reported higher Time 1 anxiety. Finally, in a three-way interaction between Time 1 levels of loneliness, day, and year of the study, we found that depressive symptoms ($\beta = 0.03$, $t$ [df = 1293] = 3.13, $p = .002$) and depressed affect ($\beta = 0.02$, $t$ [df = 2649] = 2.73, $p < .01$) increased more quickly in 2020 than 2019 for those who were higher in loneliness. Thus, depressed affect and several other forms of distress intensified at a faster rate in 2020 than in 2019, for those individuals who had high indications of this distress at the initial assessment. These findings should be replicated, however, as results were not hypothesized a-priori.

## Coping comparisons during Spring 2020 term

In the between-person analyses of coping responses during the Spring 2020 term, greater use of problem-focused coping was associated with *fewer* depressive symptoms over time

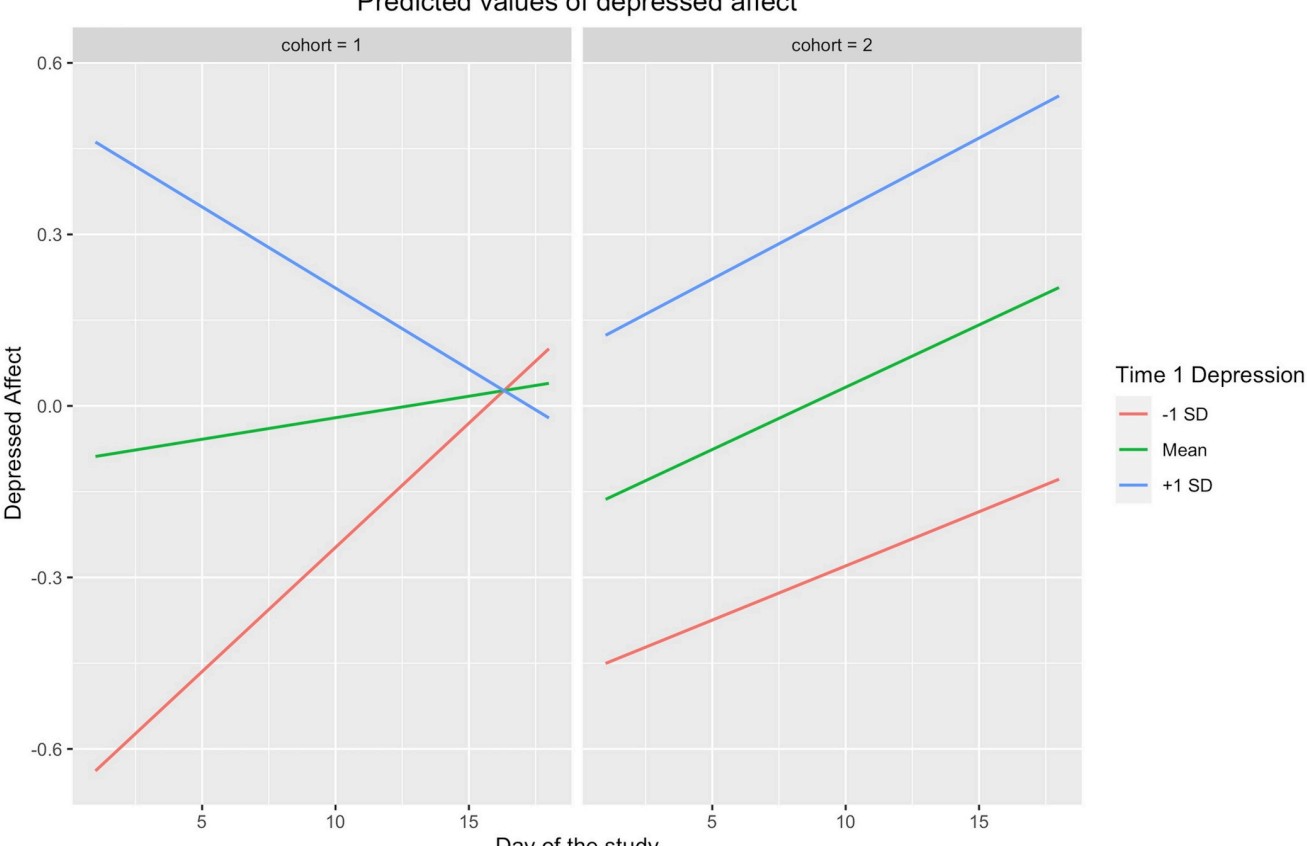

**Fig 1. Predicted values of depressed affect from Year × Day × Time 1 BDI-II interaction.** Cohort 1 = 2019; cohort 2 = 2020.

(b = -0.47, $t$ [df = 144] = -2.44, $p$ = .02; Fig 2) and *more* severe stress (b = 0.39, $t$ [df = 143] = 5.14, $p < .001$). Problem-focused coping use was not associated with loneliness ratings over time. These results are reported in Table 2.

For anxious affect, there was only a within-person effect (b = 0.03, $t$ [df = 2258] = 2.29, $p$ = .02), suggesting that participants used more problem-focused strategies when they felt anxious. For stress severity, there were both within-person (b = 0.22, $t$ [df = 2215] = 12.04, $p < .001$; Fig 3) and between-person (b = 0.39, $t$ [df = 143] = 5.14, $p < .001$) associations with problem-focused coping. Those associations suggest that participants used more problem-focused forms of coping when stressed and those who used more problem-focused coping in general reported higher levels of stress than others. Within-person problem-focused coping was associated with greater academic stress (b = 0.98, $z$ = 12.90, $p < .001$; OR = 2.66). However, problem-focused coping was perceived as more successful than disengagement-oriented coping, both within- (b = 0.38, $t$ [df = 2214] = 18.77, $p < .001$) and between-person (b = 0.84, $t$ [df = 143] = 11.82, $p < .001$).

In summary, students used more problem-focused strategies when they encountered more stress and anxiety and, on average, students perceived their coping to be more effective when they used more problem-focused forms of coping. Students who used primarily problem-focused forms of coping over time rated their coping as more effective than students who used primarily disengagement forms of coping. Note, however, that those who coped actively reported more severe stress than those who coped through disengagement.

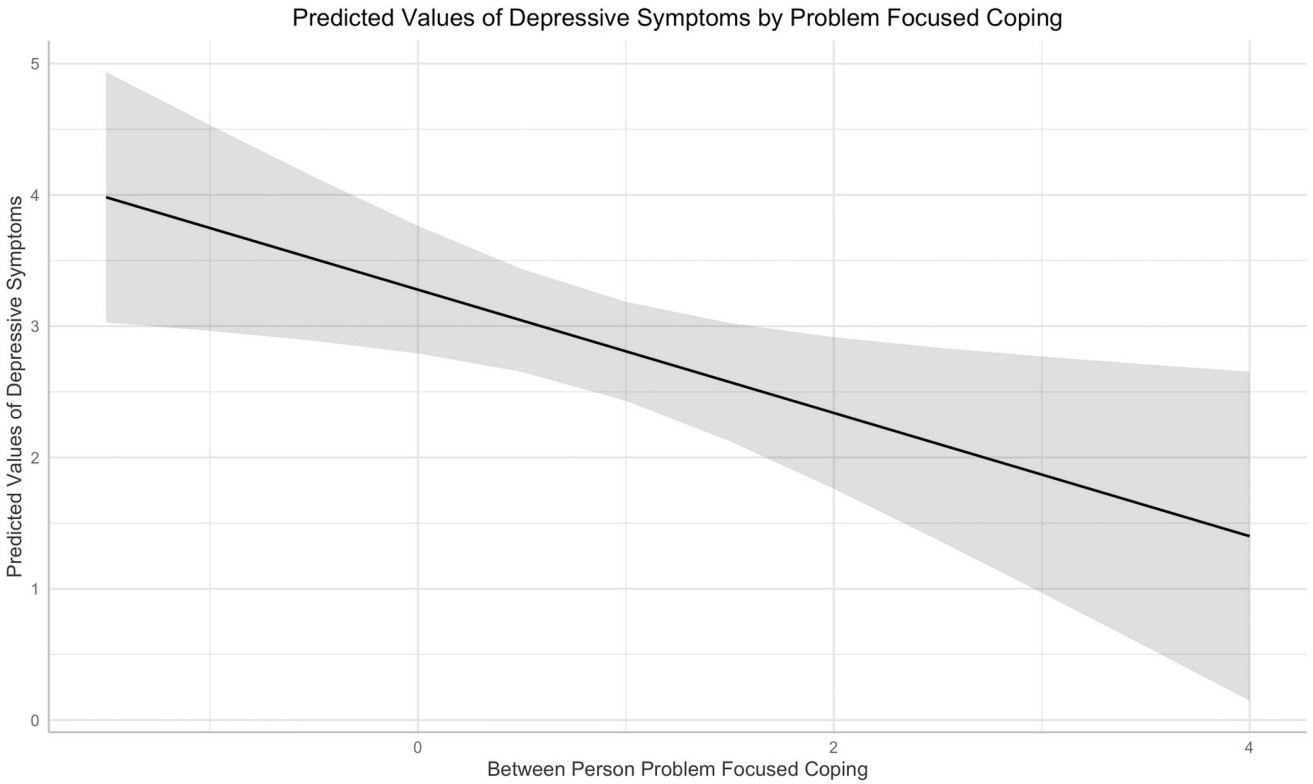

**Fig 2. Predicted values of depressive symptoms across the range of between-person problem-focused coping ratings (x-axis).** The grand mean centered coping variable, which ranged from -2 to 4, represents an individual's relative position to the overall sample in their use of problem-focused coping over the course of the term.

## Heterogeneity in responses

The mixed findings reported above called for further interrogation of the data. To recap, there were no differences when comparing 2019 and 2020 mean levels of Time 1 or Time 2 distress. However, distress indicators slightly increased over the course of the 2020 Spring term. Both of these findings relied on mean levels, which may obscure meaningful variability if there is significant heterogeneity among individuals. For additional perspective on these results, we conducted heterogeneity analyses aimed at characterizing the variability present in our data.

To start, we plotted mean and individual time trends (Fig 4). Levene's test [31], which examines the homogeneity of variance assumption in the residuals, was then used to check for equal departure from mean levels between individuals. Results indicated significant heterogeneity among participants in most models examined (i.e., anxiety [$F$ [df = 145] = 3.99, $p <$ .001]; depressed affect [$F$ [df = 145] = 3.70, $p <$ .001]; loneliness [$F$ [df = 145] = 3.06, $p <$ .001]; and composite negative affect [$F$ [df = 145] = 3.88, $p <$ .001), with some participants improving over time, some getting worse, and some remaining relatively stable. As Fig 4 illustrates, whole sample mean levels of anxiety can be misleading given the large range of ratings within and across time periods. The same can be said for depressed affect and loneliness. This heterogeneity highlights the need to look beyond mean differences as we try to understand the impact of the pandemic and remote education on students. To this end, we share thematic analysis of interviews, which characterize students' individual experiences and describe variations in their struggles and strategies.

**Table 2. Psychosocial outcomes predicted by coping strategies.**

| | **Depressive Symptoms** | | | | **Stress** | | | |
|---|---|---|---|---|---|---|---|---|
| | b | 95% CI | t (df) | p-value | b | 95% CI | t (df) | p-value |
| Intercept | 2.72 | 2.25–3.18 | 11.37 (1054) | < .001* | 6.47 | 5.95–6.99 | 24.42 (1054) | < .001* |
| Linear growth | 0.07 | 0.04–0.09 | 4.60 (1054) | < .001* | 0.04 | 0.01–0.07 | 2.73 (1054) | < .001* |
| Within-Person Coping | -0.05 | -0.14–0.04 | -1.03 (1054) | .30 | -0.05 | -0.14–0.04 | -1.15 (1054) | .25 |
| Between-Person Coping | -0.47 | -0.85 – -0.09 | -2.44 (144) | .02* | -0.58 | -0.98–0.58 | -2.84 (143) | < .01* |
| | **Depressed Affect** | | | | **Anxious Affect** | | | |
| | b | 95% CI | t (df) | p-value | b | 95% CI | t (df) | p-value |
| Intercept | 0.35 | 0.25–0.44 | 7.18 (2258) | < .001* | 0.72 | 0.58–0.86 | 10.19 (2258) | < .001* |
| Linear growth | 0.02 | 0.01–0.02 | 4.86 (2258) | < .001* | 0.01 | 0.01–0.02 | 2.29 (2258) | < .01* |
| Within-Person Coping | 0.00 | -0.03–0.00 | -0.28 (2258) | .78 | 0.03 | 0.01–0.06 | 2.29 (2258) | .02* |
| Between-Person Coping | -0.10 | -0.20 – -0.10 | -2.05 (144) | .04† | -0.04 | -0.15–0.07 | -0.76 (144) | .45 |
| | **Stress Severity** | | | | **Coping Effectiveness** | | | |
| | b | 95% CI | t (df) | p-value | b | 95% CI | t (df) | p-value |
| Intercept | 1.53 | 1.34–1.73 | 15.49 (2215) | < .001* | 1.64 | 1.44–1.84 | 16.26 (2214) | < .001* |
| Linear growth | -0.02 | -.03 – -.01 | -3.22 (2215) | < .01* | -0.05 | -0.06 – -0.04 | -7.18 (2214) | < .001* |
| Within-Person Coping | 0.22 | 0.18–0.26 | 12.04 (2215) | < .001* | 0.38 | 0.34–0.42 | 18.77 (2214) | < .001* |
| Between-Person Coping | 0.39 | 0.24–0.55 | 5.14 (143) | < .001* | 0.84 | 0.70–0.98 | 11.82 (143) | < .001* |

Notes:

† = Non-significant after applying Benjamini-Hochberg procedure.

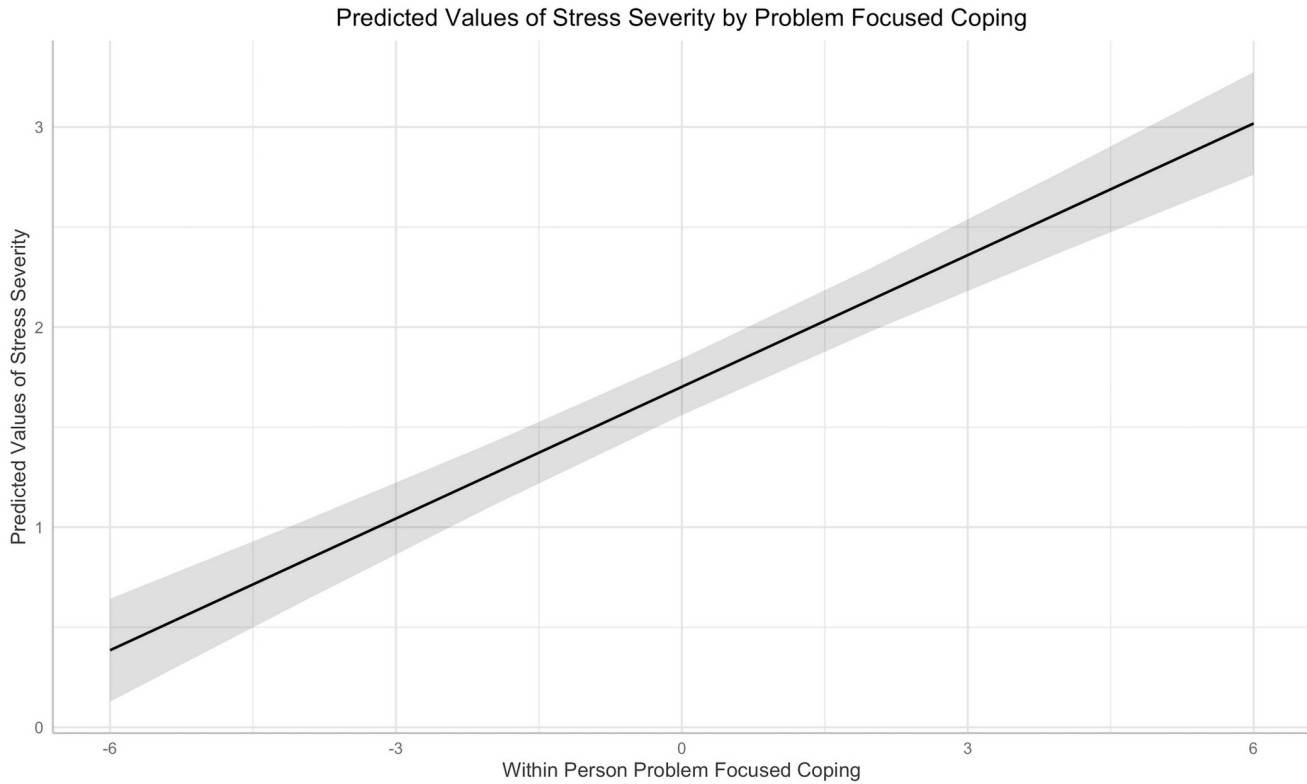

**Fig 3. Predicting stress severity from fluctuations in problem-focused coping.** The centered within-person coping variable ranged from -5 to 6, reflecting an individuals' deviation from their typical use of problem-focused coping strategies.

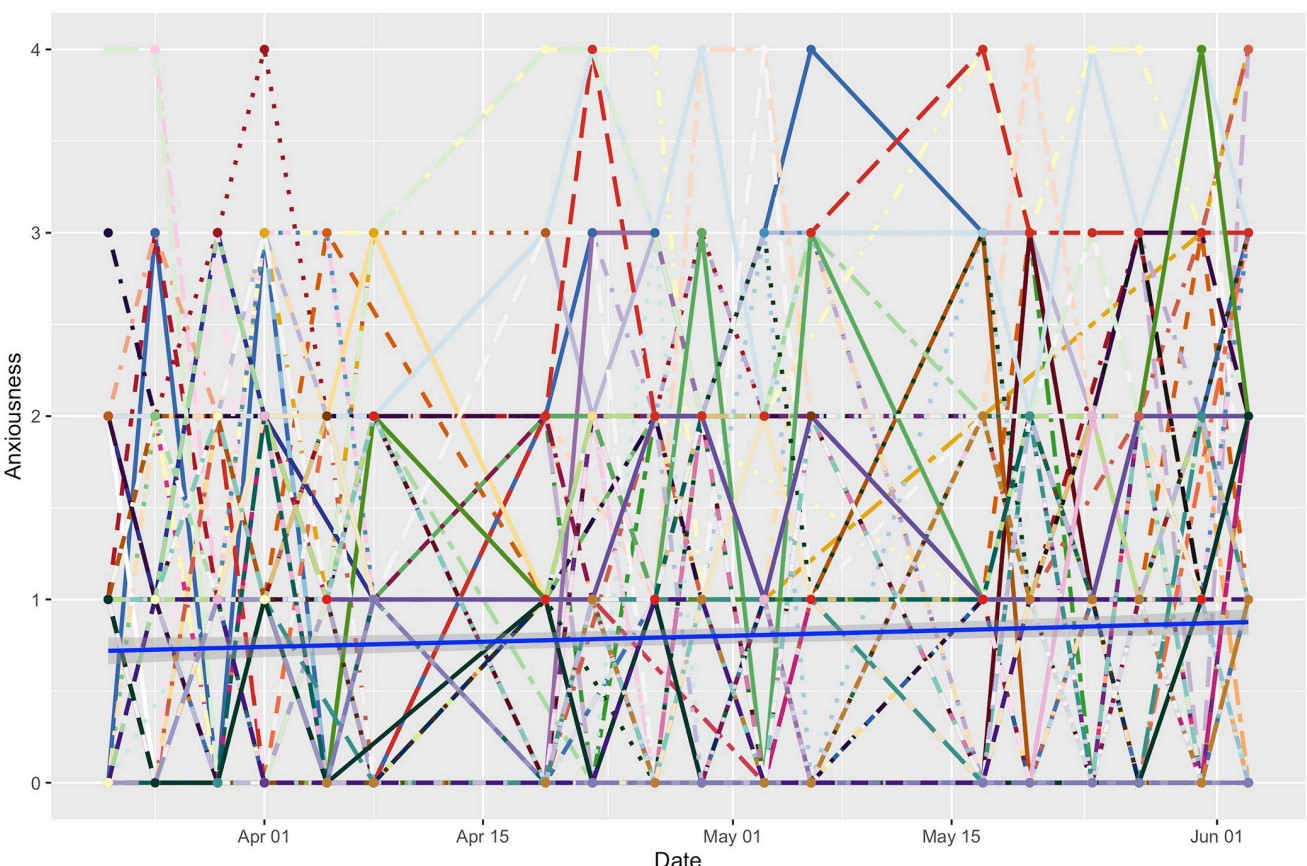

**Fig 4. Heterogeneity in individuals' levels of anxiety (reported in ESM).** Individual trajectories of anxiety are shown in different line types and colors (dotted versus solid lines represent different participants). Although the mean level of anxiety is 1 on a scale of 0–4, the significant variation in responses invites examination of individuals and subgroups.

## Qualitative results

Themes from interviews are organized below into *challenges* participants faced during the Spring 2020 term as a result of participating in college remotely from home and their *strategies* for remaining academically and socially engaged. Note that filler words and phrases that could identify participants were deleted from quotes.

### Challenges

As participants returned to their families' homes, or in several cases of commuting participants, stopped leaving home for classes and college activities, they struggled with myriad interconnected challenges. These included academic disengagement, displacement from the supportive milieu of campus, family needs that interfered with studying, restrictions to newfound independence, and difficulty meeting new people and sustaining recently formed friendships. Psychosocial challenges were often interwoven with academic challenges.

**Disengagement from school.**   Participants described a sense of detachment from college that they attributed to lack of interactivity with other students and faculty, disengagement of peers in breakout groups, and fatigue from staring at a screen for long periods. Some said they felt as if they were part-time students (even when they had a full course load) and that attending classes went from feeling like an obligation to a suggestion.

Participants had far fewer opportunities to interact with faculty and other students this term, an important barrier to academic engagement. Pre-recorded lectures were described as particularly unengaging. Additionally, the ongoing availability of pre-recorded lectures made it easier to procrastinate and watch many lectures right before an exam. Participants complained of disengagement during live lectures (held over Zoom video conferencing) as well and mentioned that the norm of students turning off their cameras made them prone to distraction and removed a sense of accountability:

"*[If] someone is watching me. . . I know that I have to be accountable. [But]. . . online learning hasn't been as effective as it should have been. Because [when the camera is off] there's no way really for [teaching staff] to know you're present and you're actively participating in class and like taking notes.*"

(P16)

Disengagement of other students during group exercises lowered participants' motivation. During breakout groups, many of their peers turned off microphones and cameras and refused to participate. Some described increased engagement when groups had to work towards an assignment of some sort rather than simply discuss a topic.

Experiences with group assignments varied, depending in part on participants' family situations and academic motivation. One participant who was caught between the demands of her family and school complained that her teammates wanted to do all of the work during their call versus working independently and then syncing. Many participants felt that they were compensating for peers who were not pulling their weight on a project, in some cases because those peers had opted for satisfactory/not satisfactory grading, an alternative to numerical grading made available to all students at this institution during the pandemic. There were also technical barriers to collaboration. One computer science major said that it was not possible to do pair programming simultaneously due to limitations with a particular platform. Even team projects that had no major technical barriers or conflicts were generally described as handoffs rather than collaborations.

The experience with online learning depended on the subject matter, with some topics and formats translating to an online format better than others. Lab classes became abstract and much less informative in their online versions. One student described his frustration:

"*I like to build things and stuff like that. I like to see it in person and feel it. So the fact that everything was online. . . . I'm just basically reading all the time. I just couldn't learn that way*"

(P19)

Similarly, class discussions were diminished when classes went online, removing opportunities for active participation.

**Displacement from shared learning environments.** Displaced from campus, some participants found it hard to get into the mindset of school and harder to connect with peers. Places on campus and the presence of peers triggered a level of focus that was harder to call up in one's bedroom or dining room. Participants spoke longingly of third places [32], such as a table in a dorm or spot in the library, where they would previously gather with classmates for study groups in an organic, unplanned way.

The family homes that participants returned to in the pandemic, particularly those of low SES participants, often interfered with learning. Participants spoke of working while in bed or at a desk crammed next to their bed, or in the dining room as family members frequently

walked through, sometimes struggling to ignore noise from others in the household. There were important exceptions: a participant with a disability was able to focus better when she did not have to struggle with walking around campus.

Many participants had insufficient internet connections—with families unable to afford upgrades. Participants coped by going to nearby relatives' homes for classes and negotiating online time with other household members. Technical support, even when well-intentioned, was not always helpful. One participant who alerted her professor about internet connection difficulties in breakout sessions was told to use Zoom on her computer for video access and her phone for voice participation—a solution that resulted in her face appearing in one break-out session and her voice in another. Another participant, housed in a tent outside her parents' small home, could only access Wi-Fi when she was very close to her parents' home, but this proximity interfered with her ability to focus and to have private conversations. Connectivity issues and other technical barriers, such as lack of access to the touch screen devices required for some types of remote collaboration (e.g., collective problem solving with an instructor's interactive white board) were expressed more commonly among low SES participants than others.

**Family needs pull students away from school.**   Family demands and cultural expectations limited some participants' ability to prioritize education. For example, a participant who was the first in her family to attend college, was often urged by her mother to do housework and help younger siblings. She felt rushed by her mother's frequent inquiries about whether she was finished with her homework and couldn't immerse herself in her work as she had done while living on campus. Another participant's grades suffered after his mother interrupted a timed quiz, requesting his help. Some participants were called on to help parents in practical ways (e.g., one participant spent some of finals week completing financial assistance forms for her parents who did not speak English). Another participant had to retreat to her room, which had poor Wi-Fi, to escape the sounds of her father talking loudly on conference calls or pounding on a treadmill. After a peak of distraction—when her parents and siblings yelled in an adjacent room as she took a midterm exam—she moved out of the family home back to a shared apartment near school.

**Autonomy interrupted.**   Autonomy was constrained in a variety of ways. Some participants felt "trapped" or that they were stepping back developmentally as they moved back into their family's homes. They felt that they lost the autonomy they had obtained when they left for college. One participant summarizes the experience: *"It was challenging . . . being independent and then being pushed back home. It's a huge change because now you have more rules again"* (P19). Several participants spoke of difficult "power dynamics," in which their parents restricted them from leaving the home due to fears of exposure to COVID-19. To have more freedom, three participants moved back to shared residences near campus after short stints in their families' homes. Other participants described more subtle barriers to autonomy, for example, not being able to have a conversation outside a parent's earshot or parents routinely entering the participant's bedroom without knocking.

**Difficulty forming and sustaining peer relationships.**   Most participants described a weakening of relationships with new or potential friends from college. This theme appeared in every interview with low SES participants and in almost all other participants. In contrast, communication often intensified with immediate family, high school friends and well-established college friends.

Participants found it difficult to form new relationships with classmates and to sustain relationships with classmates they did not know well for academic support, friendship or dating. The lack of shared experiences and shared physical spaces made it harder to strike up informal conversations with classmates. One participant said that the easiest way to make friends at

school was to mention to someone, either right after class or in another context, that *"we are in this class together. . . then you get their number and form a study group"* (P05). There was no natural situation to make these connections online, raising the bar for conversing with someone who was not already a friend. Another participant missed the opportunities for casual conversation in her sorority house: *"In a sorority, whenever I got bored I would just go down to a common space. And usually other people would be there and I could strike up a conversation with them"* (P24). One participant noted the loss of incidental contact that he previously had when walking around his department and asking other students what they were working on. Be it a specific department building, residence or the campus as a whole, the shared physical spaces signified shared experiences and permitted easy conversations and feelings of affiliation. This social scaffolding was lost during the pandemic.

The lack of interaction with peers also made it harder to establish study partners. Participants who had not established a person or group to study with before the start of the Spring 2020 term had difficulty doing so remotely. They found it harder to assess compatibility without seeing how engaged peers were in class. Some of these students felt that not having study partners hurt them academically. In previous terms, students were motivated not just by study partners but also by roommates and peers working in a shared space. Participants contrasted the positive influence of their peers with the distraction of family members.

One barrier to peer interaction was a lack of transitions from formal educational situations to informal interactions: classes and meetings had a hard stop rather than transitioning to a walk across campus or hanging out in a dorm. One participant contrasted the organic flow from study sessions to socializing she had enjoyed in previous terms to the time boxed interactions she had during the pandemic *"we're probably just hanging out with each other and you know like get something to eat after, get something to drink. . .. Now we just hang up and go about your day"* (P08). One participant said that whereas her Friday game sessions with friends formerly extended into weekend meetups, the follow-up hangout sessions didn't happen during the pandemic. Groups involving hands-on creativity (e.g., music collaboration, dance clubs, robotics groups) were less compelling online and stopped meeting early in the pandemic.

**Cycles of psychosocial and learning challenges.** Mental health concerns were top of mind for participants. Many shared variants of the general statement *"my mental health took a big hit"* (P19). All but two participants described challenges to their general wellbeing such as disrupted sleep patterns, decreased motivation and a lack of routines and structure. Further, 16 of the 27 participants described feeling depressed or anxious. These psychosocial challenges were intertwined with academic challenges—dynamics that changed over the course of the term. For example, anxiety initially motivated one participant, but that motivation decreased as the term progressed. Another participant was initially more focused at his family's home than he had been at school because there were fewer distractions. But as the term progressed, feelings of loneliness impaired his focus.

Several bi-directional links between psychosocial and academic stress were described. One link was between depressed mood and academic disengagement. A participant with a history of depression said that her mood worsened over the Spring term, in part due to the detachment she felt in online learning. Lowered mood in turn made her feel even more detached from her classes. Others described a link between academic performance and anxiety. Worries about grades and long-term goals made one participant so distressed that he didn't leave his bed for weeks—a withdrawal which itself threatens the ability to engage with classes. He described intense worry about his grades, which suffered during the Spring term:

"*Just like helpless. I felt terrible because I did not think I was going to do how I ended up doing in my classes. I was so worried I was going to fail. I was so worried I wasn't gonna be allowed to stay in my [department] because you have to maintain a certain [grade point average] and I was so worried I wasn't going to get my major because of my class grades and how things were going. . . . Um yeah I was just in a hole for a while. . . . My academic stresses. . . it was too much at one point for me. I was just at a very low point. I was just bedridden for like two to three weeks.*"

(P19)

This distress also interfered with sleep, which may have added to his academic challenges. Many students, like the one above, described disrupted sleep routines that made it harder to focus or sometimes even to attend classes. They attributed sleep problems to a number of factors including worries about school, the lack of physical boundaries between a workspace and bed, and shifts in routines that came as a result of living with family. One participant described insomnia and worries about falling behind as a result:

"*I mean I'd say my biggest mental health challenge. . . just like general insomnia has been pretty hard for me. Especially because I would spend a lot of time lying in bed not doing anything when I had a lot of homework to do the next day. So then I would become stressed about whether I'll be able to finish that homework or not. And it was just frustrating because I would end up losing huge chunks of time.*"

(P04).

Another participant knew that her sleep had become seriously disordered and that this was taking a toll on her attention and her health, but she wasn't sure how to get back into normal routines without falling further behind. She thought it was better to wait until the end of the term to either seek help or add more structure to her days. She and several other low SES participants mentioned uncertainty about how to access treatment for mental health concerns or long waiting periods for treatment. Some participants were comfortable with teletherapy and felt it was beneficial, but others lacked the acoustic privacy in their homes to have private conversations (i.e., to speak openly about personal struggles without being overheard). Participants had not heard of text-based therapies or conversational agents, but expressed interest when these tools were explained in interviews. Apps for relaxation had value for several participants. Some participants felt that they had to resolve problems on their own.

## Strategies

As classes went online in the Spring term, participants tried to stay academically engaged and to maintain social connectedness. The six themes below relate to one or both of these overarching goals. Common threads such as initiative-taking and skillful use of communication technologies cut across the themes.

**Self-learning and activation.** Some students described an increased independence in the way they learned over the course of the term. Participants expressed pride in self-learning rather than relying exclusively on lectures and assigned materials. One participant explained:

"*For a few of my classes I feel like actually [I] was self-learning because sometimes it's hard to sit through hours of lectures and watch it. It might just be easier to Google the equation and figure out how it works instead of listening through all of it. And so I didn't watch all of my*

*lectures. Sometimes I just use Khan Academy or something and self-learning. So this quarter a lot more self-learning happened than in previous quarters.*"

*(P06)*

Even those who merely read additional material or worked on additional problems that were suggested by an instructor as optional seemed to feel greater self-efficacy as a result.

Some took learning into their own hands in more extreme ways. Frustrated with reading abstract descriptions of experiments he expected to learn hands-on in a lab, one participant took to his garage where he made up experiments of his own. This intense hands-on, self-driven learning helped him emotionally as well as educationally. Reconnecting with his curiosity and creativity in this way helped him out of an extended emotional struggle.

"*I just took apart old electronics and then I would look at the surrogate diagrams for things and I would read the lab [and say] 'Okay I think I could figure this out'. . .. I learned about the major concepts overall I think a lot better . . . and it was kind of nice seeing it as opposed to just reading it in the book. It's kind of like seeing where it's applied. . .. And then I would just stay up all night just building things. I enjoyed that. . . . I found pleasure in that and then it kind of opened up a ton of pathways for me. I was like 'Okay I can still do programming because all of my projects require it you know or I could still like do all these things.' And I was not dead set on that one major anymore.*"

(P19)

Other participants immersed themselves in creative or self-learning pursuits that were not related to their classes. Although not undertaken as therapies, these independent pursuits did have psychological benefits. One participant who had previously dabbled in drawing became extremely involved in Anime communities on Instagram, where she struck up collaborations. She felt proud of this work and the recognition she received from other artists. Another participant, wanting to contribute in a positive way during Black Lives Matter (BLM) protests, forayed into journalism by interviewing individuals from different factions at protests and blogging about it. He felt more connected to the outside world as a result. One participant shared his poetry on the user-contributed sharing and discussion platform, Reddit, and another worked on new technologies to support frontline medical workers.

**Structuring routines and environment.** Recognizing the need for more structure, many participants deliberately instituted routines, for example time-blocking specific activities. Several used physical calendars to mark assignments and timelines, finding that they forgot to look at online calendars. One participant renewed her focus by reorganizing her room. Many established new exercise routines.

To get into a school mindset, some participants tried to reduce distractions and multitasking. One participant took a phased approach, at first cutting out any screen distractions that had a verbal component (e.g., a video with dialogue) and then cutting out all multitasking while listening to lectures. This participant and others moved their phones out of reach during lectures. To stay focused, many students watched recorded lectures at 1.5 or 2x speed and forced themselves to take notes during lectures. Some participants felt that watching recordings of live lectures at times that better fit their new schedules, rather than watching them in real time, allowed them to optimize their time. For participants observing Ramadan, which took place during the Spring term from April 12 to May 11, the ability to watch early morning lectures later in the day allowed them to adapt their sleep to fasting schedules.

To manage family interruptions, participants either communicated their schedules to family members (verbally, with a physical calendar, or sign) or adjusted their own reactions to interruptions. For some participants, it was important to think about boundary setting in a more general way as they went home and were asked to play caregiving roles. Therapy and a podcast about demands on women of color helped one participant focus on her academic priorities and recognize that they conflicted with her family's requests. To help with emotional wellbeing, some participants deliberately reduced their intake of media related to the pandemic and BLM protests as the term progressed.

**Learning with peers.**   Remote study groups offered crucial support for many participants. These study groups, held through various video and voice communication platforms (e.g., Zoom, Discord, FaceTime) and sometimes just a messaging thread, were typically analogues of meetings that formerly occurred in person. Study groups were often subgroups of students in the same class or cohort (e.g., a subgroup of students in the first year of an engineering major). One participant, grateful that she was invited to join a study group by a student she didn't know beforehand, thought the group allowed her to succeed in a challenging class where she otherwise would have struggled. Peer support groups, set up by students for anyone in the class to ask or answer questions by messaging or voice, were also helpful, especially for those participants who were not part of other study groups.

Informal coworking sessions, for example, video calls among friends or peers who were working on assignments for different classes, helped participants to stay on task and give each other moral support. These coworking calls varied in length from one to eight hours and were interspersed with consultation and personal conversation. In addition to peer consultation and accountability, calls held over FaceTime and WhatsApp kept participants focused by tying up their phones—which could otherwise be a major source of distraction.

**Participating more online.**   Several participants asked questions in large lectures for the first time, finding it less daunting to ask questions online than raising their hands in a lecture hall. In the physical lecture hall, these participants, who identified as women, worried about being negatively judged by other students and feared that the instructor would ask them follow-up questions. One participant explained:

"*It felt easier to ask questions [in] large lectures because you're turning your [microphone] on like everyone can hear. You don't have to repeat yourself. Yeah so I think actually [I] found it was easier online to participate more because I'm not seeing everyone's faces staring at me.*"

(P21)

In previous terms, this participant avoided asking questions in large classes because she felt shy. Watching as other students' questions were misunderstood or ignored by the instructor increased her reservations. Instead of raising her hand to ask questions, she turned to the friends she sat with in class. It should be noted that participants had opposing preferences for online question asking options. While the "raise hand" and chat features on Zoom offered some students a more comfortable way of participating, they deterred others. One participant was so uncomfortable imagining other students seeing a large image of his face, a consequence of the "raise hand" feature, that he did not ask questions.

Participation in office hours and meetings with advisors was easier online. Instead of walking across campus and then waiting, students could sign up for a slot online and then share their screen with an instructor to get help with problems. Some participants who had never previously attended office hours did so this term because it was so much more efficient:

"*And I also went to office hours a lot more this quarter. Just because it's so easy to pop into a Zoom and ask a question. . .. usually someone [would ask] a question and the professor would go through that problem sharing their screen. . .. this [term], I could go to office hours until the minute before my next class started.*"

(P26)

In the example above, office hours allowed peers to hear and respond to each other's questions. This peer consultation was missed when online office hours were one-on-one meetings with instructors.

Some participants not only participated more online, but also took on leadership roles by initiating group discussions. One participant described a positive experience in a review section that required students to turn on their cameras and microphones. During the breakout groups of this review section, she took it as a challenge to engage other students in discussion:

"*And so I kind of took on the facilitation role to make sure we're getting through all the specific questions we're supposed to talk about and doing the activities. . . I would fill in my answer and then ask if anyone else had any other thoughts.*"

(P24)

These types of participation—asking questions and guiding other students—were clearly positive experiences. In these instances, the students were actively involved in online learning, rather than frustrated consumers of it.

**Emotional wellbeing through communication platforms.** While some participants found help in teletherapy and meditation apps, most participants emphasized the emotional benefit of communicating with their friends. When asked what was most important for her emotional wellbeing during the Spring term, one participant's response was typical, *"Honestly I think the access to technology and just being reassured that contacting anyone was only a couple clicks away"* (P24). For some participants, it was important to see their friends' expressions and hear their voices over calls or video, either for evidence of their friends' wellbeing or for the lift they felt in perceiving their friends' positive emotions. For others, text messaging was the primary mode of communication. It required no planning or coordination across time zones and felt private since no one could overhear the conversations. One participant noted that texting was appropriate for emotional conversations: she and her friends could still understand each other even when they were crying.

Communication with friends often involved emotional support. One participant tried to support an anxious friend by immediately responding to her texts, addressing each of the many concerns she raised point by point. He couldn't solve the problems, but tried to show her that he was paying attention. One participant wished that she could have offered face-to-face support for a friend who shared personal news:

"*A friend came out to me over text and. . . I would have rather done it in person. . . being able to take body language cues that you don't really get over text. . . I really would have liked to be able to provide a hug. . .. it would have just let me know like 'oh she's nervous about this' [or] 'she's super confident'. . .. So I know how to amp up my reaction . . . like more aggressively supportive if I knew she was super nervous about it.*"

(P14)

She gave a supportive response but felt that if she had more cues about her friend's emotional state, she could have been more confident that she was responding with the appropriate tone.

Some participants learned that by taking more social initiative than they had previously, they were helping their friends as well as themselves. One participant described,

"*In general, I'm not very good at reaching out to people, but this quarter has forced me to think about relationships and be like 'Okay I need to actually reach out, talk to them, maybe FaceTime them and see how they're doing and talk about everything that we're going through and make sure they're okay.'''*

(P12)

As described below, this initiative extended from one-on-one communication to group connection. Participants used low pressure tactics, such as notifying friends that they themselves would be online in a video call during a certain time range in case anyone wanted to join. They also made it clear that people could come and go, so there was no feeling of obligation to stay on the call for its duration.

**Using familiar technologies in new ways to communicate with friends.** Here we build on the previous theme about use of communication platforms to support emotional wellbeing. We specify how they used those tools to sustain their friendships and in turn, their own wellbeing. Participants drew on the same communication platforms they had been using before the pandemic, but used them more intensively, for different purposes and combined them in ways that they had not previously.

Video calling was used for a sense of being in the company of peers. Participants used video calling less for focused conversations than for extended coworking and hanging out. One participant described hosting coworking sessions with friends that lasted anywhere from four to eight hours:

"*I would initiate. . . we have a study group chat and every day I would be like 'Hey I'm going to be on at this time starting at this time.' So then I gave them time to all have the room open for Zoom and stuff. Okay and then any time after that they can join and then said I [would] wait like maybe 30 minutes or even an hour. . .. And then people join and then we work maybe . . . till midnight, a little bit past midnight*"

(P21)

Casual conversation was often interspersed with studying during these coworking sessions. One described intervals of 40 minutes working and 20 minutes chatting. In purely social video calls, participants talked, played games and simply hung out with friends for hours at a time. Similar to its value in long distance relationships [33], extended video calling allowed students to feel present in one another's lives.

To work around the formal and rigid nature of some video calling apps that require scheduling and communication with an entire group at once, some participants reached out to friends with impromptu video calls over FaceTime, WhatsApp, checking to see who else was on a platform in between scheduled meetings (e.g., on Discord, a platform popular for gaming and communication), and texting. A participant who hosted Discord groups for multiple clubs seeded conversation topics to invite chats outside of scheduled sessions.

In addition to the explicit emotional support discussed in the previous theme, lightweight sharing was another way that participants maintained friendships. For example, one

participant turned snippets of pre-recorded lectures played at a fast speed into memes that she shared with her friends via social media messaging. The snippets poked fun at a professor and herself for watching her pre-recorded lectures at a fast speed after procrastinating:

> "*like if my professor said something funny in the lecture and I would send a video of that over Snapchat. . ..I'd gotten used to the speed of the voice but other people would hear that video and just be like 'Wow, you must be procrastinating*!'"

(P14)

As they did before the pandemic, participants shared images with different degrees of personalization depending on closeness in a friendship. But in this time period, image sharing became the lifeblood of many relationships rather than a supplement to in-person contact. The ability to exchange pictures in ways that resonated with friends and distant contacts became more important, as did other forms of effective remote communication.

To meet new people, some participants used dating apps and others turned to large forums:

> "*Everyone was bored, everyone was doing it. You know people just want someone to talk to outside their families. . ..And I use Tinder like socially not really in terms of finding a relationship or anything but just to talk. . ..But I met people in Singapore and South Korea and Texas and California.*"

(P23)

Some narrowed their matching criteria to a shared time zone to find compatible conversation partners. Some participants who had not done so before started engaging with larger groups over social media: participation in political dialogue allowed some to meet new people and feel part of a community.

## Discussion

Previous studies on psychosocial experiences over the course of the pandemic are conflicted, with some finding no changes in mental health and others reporting marked rises in mental health concerns. Our analyses shed light on these mixed findings by showing that college students' experiences are themselves mixed. Emotional and academic distress vary over time and across students. The findings of heterogeneity resist sweeping conclusions about either the resilience or vulnerability of college students and instead suggest value in examining individual experiences.

Through thematic analysis, we offer a contextualized illustration of the psychosocial and academic distress experienced by students during the pandemic. We describe the day-to-day challenges associated with participating in college remotely and examine how students cope with these challenges. Academic and psychosocial distress were interwoven, with decreased interaction in online learning affecting both motivation and feelings of connectedness. The problems were cyclical such that feelings of isolation sometimes made it harder to stay motivated and to contact peers. Difficulty establishing study partners, in turn, made it harder to succeed in difficult classes. Similar cycles played out with other forms of psychosocial distress. One student described how academic detachment worsened her depression and vice versa. Academic worries sometimes gave rise to more general anxiety and insomnia, which, in turn, worsened focus. The particular cycles of distress varied across students, and the distress of particular students changed over the course of the term.

On the positive side, students described strategies that they felt were helpful both academically and psychosocially. In general, students described feeling better when they took initiative. Academically, this meant learning in a more independent style, structuring their routines and environments, and coordinating study sessions with peers. Socially, this meant checking in with friends to give and receive social support, lightweight bonding, and arranging hanging out and coworking sessions with peers. Students used technologies with which they were already familiar, but applied them in different ways and for different purposes than they had before, whether that was using video calling for multi-hour coworking sessions rather than quick greetings, or an app like Tinder to find new conversation partners rather than dating prospects.

Both our quantitative and qualitative results indicate that the students who show this kind of initiative, that is, those who created active coping strategies to address the problems in front of them, felt that they were coping effectively. But, in a seeming paradox, these students experienced more severe stress than students who coped by disengaging. Those who disengaged reported they experienced less severe stress overall, but also reported more symptoms of depression over time and perceived their coping to be less effective. These findings fit with models of stress coping responses that associate disengagement with higher risks for psychopathology [17]. The effect of COVID-19 related stress on an individual's mental health depends not just on the stress itself but on the particular coping strategies used by that person [18]. Our data suggest that effective coping strategies do not necessarily reduce stress, in fact they are associated with more severe stress over time, but may protect against the potential deleterious effects of that stress (e.g., depressive symptoms). Stressors associated with the pandemic and remote learning persisted regardless of what coping mechanisms were applied: throughout the term, classes offered little interaction, socializing had to be scheduled, and the needs of family members continued to compete with school. Some students acknowledged and directly worked on these challenges and others looked the other way. Neither approach made the challenges go away, but we see from interviews that students benefited when they acknowledged challenges and created problem-focused strategies that were tailored to their social and technical contexts.

Those who are developing, planning or deploying psychosocial interventions to support students should consider the varied experiences within and across individuals. Our data suggests that some students, in particular those with psychosocial vulnerabilities, experienced more difficulties than other students during the pandemic. In interviews, some students described either a rekindling or intensification of mental health concerns during the Spring term. In addition, those who reported more distress in initial survey assessments continued to experience more distress than other students, with distress increasing at a faster pace in the Spring of 2020 compared to the previous year. The difficulties encountered by low SES students were highlighted in thematic analyses of interviews. Accounts of family distractions, technical barriers, and disconnection from college peers experienced by low SES students echo past research on socioeconomic disparities among college students during the pandemic [16].

Variations among these vulnerable students are also important to note. They experience distress at different times and in response to different challenges. They have differing preferences regarding psychosocial support. Some wanted to solve problems on their own, some were comfortable with teletherapy and others wanted more discrete forms of help. Academically too, students had different preferences for support. Even though most students wanted to feel comfortable participating in classes and office hours, they had conflicting preferences for the online features that might enable this engagement. Online features for asking questions emboldened some students to ask questions for the first time this term but raised social anxiety for others. Online office hours were similarly polarizing, with some appreciating the

convenience and others craving the peer consultation that was a natural part of in-person office hours.

Rather than relying on a one-size-fits-all approach to supporting students who are participating in college remotely, our findings of heterogeneity underscore the importance of tailoring interventions, keeping in mind the context for individuals' struggles. Past research has established the general importance of problem-focused skills in reducing psychosocial distress [34]. This paper situates problem-focused coping within the daily lives of young adults who are reliant on communication technologies for learning and social connection, providing examples of skills that were helpful in the context of the pandemic. To help students develop effective coping skills, the skills training that is commonly part of cognitive behavioral and dialectical behavioral therapies could be offered to all students, not just those in treatment for mental health concerns. Mobile therapy apps have illustrated how access to these offerings can be extended [35–38]. However, such skills training should be tailored to the specific contexts of students and the tools that are woven into their social and academic lives. Given that the specific challenges facing students, along with accepted communication platforms, are continually changing, peer role modeling may be more appropriate than expert training, or at least play an important role. Students may learn from peers who are adept in using communication technologies to manage current learning and relationship challenges.

## Limitations

This exploratory study of college student experiences during the pandemic was limited by the self-report nature of assessments and time-based measures as a proxy for pandemic effects. To understand experiences of distress and coping strategies we relied on students' responses to online surveys and on their accounts shared in interviews. Students' evaluation of their effectiveness in coping was not benchmarked with objective measures such as grade point average. We decided that grades would be unhelpful as a benchmark in Spring 2020 because of policies such as optional exams and the satisfactory/not satisfactory versus numerical grading that were offered to students.

Time-based comparisons served as a proxy for pandemic effects. We inferred pandemic effects by comparing survey and ESM responses from 2019 to those in 2020; a subset of which was a within-person comparison of students who participated in multiple years of a study. Historical events other than the pandemic may have affected ratings. The year-to-year between-person comparisons are limited by differences among students. These students were not matched by year in school or area of study. The comparison of slopes from 2019 to 2020 to assess rates of increasing distress was not hypothesized a priori and needs to be replicated by other studies. Finally, we limit our conclusions to college students. We recognize that findings about effective coping may not apply beyond this relatively high functioning sample.

Future work can address some of these limitations. For example, future studies could draw on data from communication platforms for objective measures of social connectedness. In addition, analyses of the interaction between year, day, and Time 1 vulnerability should be replicated.

## Conclusion

This paper describes the challenges undergraduate students faced as they participated in college remotely during the COVID-19 pandemic. Our findings shed light on disagreements in past research about whether there is a mental health crisis among young adults due to the pandemic or if they have been unscathed. Neither is accurate, of course. We see a continuum of distress that varies across students and over time. This contextualized inquiry shows the factors

in daily life that contribute to psychosocial and academic distress, and how the two forms of distress are intertwined. We see that students benefit by creating their own active coping strategies, tailored to their particular social, academic and technical contexts. These active coping strategies, for taking initiative in learning and peer communication, can inform interventions to help students at greatest psychosocial and academic risk. These interventions are critical for supporting students when classes are fully online and may and may have value in the gradual return to hybrid and in-person models of instruction. Additional channels for participation, whether for asking questions in class or engaging in group study, will benefit students who continue attending college remotely as well as those who return to in-person education after the pandemic.

## Supporting information

**S1 Table. Participant demographics.**
(DOCX)

**S2 Table. 2019 and 2020 ESM methods and compliance.**
(DOCX)

**S3 Table. 2019 and 2020 comparisons.**
(DOCX)

**S4 Table. Time 1 and Time 2 (2020) comparisons.**
(DOCX)

**S5 Table. Differences in 2020 ESM reports over time.**
(DOCX)

**S6 Table. Differences in pooled ESM reports based on Time 1 depression scores.**
(DOCX)

**S7 Table. Differences in pooled ESM reports based on Time 1 anxiety scores.**
(DOCX)

**S8 Table. Differences in pooled ESM reports based on Time 1 loneliness scores.**
(DOCX)

**S9 Table. Multi-level models of Year, Day, and Time 1 depression interactions.**
(DOCX)

**S10 Table. Multi-level models of Year, Day, and Time 1 anxiety interactions.**
(DOCX)

**S11 Table. Multi-level models of Year, Day, Time 1 loneliness interactions.**
(DOCX)

## Acknowledgments

We thank the participants who shared their experiences for this study.

## Author Contributions

**Conceptualization:** Margaret E. Morris, Paula S. Nurius, Yasaman S. Sefidgar, Eve A. Riskin, Anind K. Dey, Sunny Consolvo, Jennifer C. Mankoff.

**Data curation:** Margaret E. Morris, Kevin S. Kuehn, Jennifer Brown, Han Zhang, Yasaman S. Sefidgar.

**Formal analysis:** Margaret E. Morris, Kevin S. Kuehn, Jennifer Brown, Paula S. Nurius.

**Funding acquisition:** Eve A. Riskin, Anind K. Dey, Jennifer C. Mankoff.

**Investigation:** Margaret E. Morris, Kevin S. Kuehn, Jennifer Brown, Xuhai Xu, Jennifer C. Mankoff.

**Methodology:** Margaret E. Morris, Kevin S. Kuehn, Paula S. Nurius, Sunny Consolvo, Jennifer C. Mankoff.

**Project administration:** Margaret E. Morris, Jennifer C. Mankoff.

**Resources:** Margaret E. Morris, Paula S. Nurius, Eve A. Riskin, Anind K. Dey, Sunny Consolvo, Jennifer C. Mankoff.

**Software:** Kevin S. Kuehn, Yasaman S. Sefidgar, Xuhai Xu.

**Supervision:** Margaret E. Morris, Jennifer C. Mankoff.

**Validation:** Margaret E. Morris, Kevin S. Kuehn, Jennifer Brown, Paula S. Nurius.

**Visualization:** Kevin S. Kuehn.

**Writing – original draft:** Margaret E. Morris, Kevin S. Kuehn.

**Writing – review & editing:** Margaret E. Morris, Kevin S. Kuehn, Jennifer Brown, Paula S. Nurius, Han Zhang, Yasaman S. Sefidgar, Xuhai Xu, Eve A. Riskin, Anind K. Dey, Sunny Consolvo, Jennifer C. Mankoff.

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
