## [Decision Letter · Decision Letter 0]

6 Mar 2021

PONE-D-20-34654

College from home during COVID-19: A mixed methods study of heterogeneous experiences

PLOS ONE

Dear Dr. Morris,

Thank you for submitting your manuscript to PLOS ONE. After careful consideration, we feel that it has merit but does not fully meet PLOS ONE’s publication criteria as it currently stands. Therefore, we invite you to submit a revised version of the manuscript that addresses the points raised during the review process.

I would particularly suggest you consider the comments provided by Reviewer 2 regarding Time 2 comparison and interview protocols and data analysis.

We look forward to receiving your revised manuscript.

Kind regards,

Amanda A. Webster

Academic Editor

PLOS ONE

Journal Requirements:

2.Thank you for stating the following in the Financial Disclosure section:

"This project was funded by National Science Foundation grants EDA-2009977 awarded to JM, PN, AD, and ER, CHS-2016365, and CHS-1941537, awarded to JM. Funding was also provided through a research gift from Google awarded to JM and a grant from Samsung awarded to JM, PN, AD, and ER.

KSK (the second author) was supported by a National Research Service Award from the National Institute of Mental Health (F31MH117827)."

We note that one or more of the authors are employed by a commercial company: Google

We note that you received funding from a commercial source: Samsung

3.We note that the grant information you provided in the ‘Funding Information’ and ‘Financial Disclosure’ sections do not match.

Reviewers' comments:

Reviewer's Responses to Questions

**Comments to the Author**

1. Is the manuscript technically sound, and do the data support the conclusions?

Reviewer #1: Yes

Reviewer #2: Partly

2. Has the statistical analysis been performed appropriately and rigorously? 

Reviewer #1: Yes

Reviewer #2: I Don't Know

3. Have the authors made all data underlying the findings in their manuscript fully available?

Reviewer #1: Yes

Reviewer #2: No

4. Is the manuscript presented in an intelligible fashion and written in standard English?

Reviewer #1: Yes

Reviewer #2: Yes

5. Review Comments to the Author

Reviewer #1: This is a very relevant study, well-designed and clearly written.

I agree with the methodological decisions of the authors and I find it valuable for other researchers in the field, since the findings underline the need for interventions oriented towards problem-focused coping, and suggest opportunities for peer role modeling.

Reviewer #2: The study uses an interesting methodology to sample data in snapshots collected regularly via text message, so data gathered at regular intervals across a couple of years. That the study has been under way since before the pandemic provided an opportunity to gather data on students’ experiences. There is obviously a very rich data set here.

I have a concern about Time2 (March 2020) as the key point for comparison. This was very early in the pandemic, particularly in the US, where lockdowns and other measures were only just beginning. A justification for this early view would be helpful as motivation for this study. The earliest comparisons suggest that some students were less-stressed in March 2020 compared to the earlier comparison points. Could this be a novelty effect? Was this somehow accounted for? Does the claim about low-stress hold up given the earliness of the data gathering? The links to coping strategies are important for us as educators to consider how to best support students through difficult circumstances.

I note that, for international readers of this paper, ‘spring’ and ‘autumn’ reference Northern Hemisphere seasons and this confusion could be obviated by using months instead of seasonal references. It would also be helpful to include a statement about how the local area and university were impacted and by what timeline as the pandemic grew and public health measures came into force. What were these? How did your university respond? This local context is important for considering the results and implications of this study. Further, do students take year-long courses at your university? Or all of them semester-long? Or a combination? If year-long, students may have had time to develop relationships, but if semester-long, the absence of personal interactions may be more acutely felt.

This study is embedded in a larger one, which gives a good grounding for asking the questions in this study. The analyses provide reasonable means to answering them. The statistical tools offer fine-grained and intensive data about the effects on and experiences for undergraduate students through the pandemic. The statistical analysis in the earlier section is detailed and reasonable, and the two sets of framing offer reasonable grounds to interpret the stats through the interviews.

I think limitations for aggregating big data needs to be acknowledged. There were also differences between Likert Scale ratings, and thus these would be interpreted differently. Similarly, self-report data is often called into question. How were these issues dealt with when the results were interrogated?

A growing literature base demonstrates and details challenges undergraduates faced during 2020/covid. Some of these are local-level challenges faced by students and the value of this study is the fine-grained detail that highlights and shows comparisons with earlier years. The study reports themes/challenges that may be common with what other research has reported, but the depth present here adds to this literature with its fine-grained detail.

I’d like to know more about why authors chose to organize themes from interviews as challenges and strategies, even as this is presumably a choice about how to present results.

There will be further interesting responses to the implications of the pandemic at the university level, particularly as many universities are pushing toward more online work while reorganizing/restructuring in response to financial strains.

There are important messages in the interview analysis, especially around social interactions that are afforded by on campus/in person activity that is necessarily associated with being a college student. This sort of engagement is perhaps underappreciated by those who advocate online learning for efficiency, cost etc, which is another key important result of the reported research—to draw attention to this aspect of college/university life and the social/protective factors entailed.

Where is the interview protocol?

p. 38, line 847 says there was training offered in CBT. The study didn’t talk about training.

The final discussion points do not really follow from the paper and what was presented.

There are very few minor production errors, misplaced apostrophes, etc

6. PLOS authors have the option to publish the peer review history of their article (what does this mean?). If published, this will include your full peer review and any attached files.

Reviewer #1: **Yes: **Dimitrios Vlachopoulos

Reviewer #2: No

---

## [Author Response · Author response to Decision Letter 0]

23 Mar 2021

PONE-D-20-34654

College from home during COVID-19: A mixed-methods study of heterogeneous experiences

PLOS ONE

Thank you for the opportunity to revise and resubmit our manuscript. We appreciate the thoughtful review of this work. Below, we address the concerns raised in the review. 

Editorial issues and disclosures (grants, funding roles) 

D1. Formatting 

We have reformatted the main body of the manuscript and the tables to comply with PLOS ONE guidelines. 

D2.a. Amended financial disclosures 

In the cover letter, we include an updated funding statement clarifying commercial involvement (below).

“This material is based upon work supported by the National Science Foundation under Grant Nos. EDA-2009977 awarded to JM, PN, AD, and ER, CHS-2016365, and CHS-1941537, awarded to JM. Funding was also provided through a grant from Samsung awarded to JM, PN, AD, and ER.

KSK (the second author) was supported by a National Research Service Award from the National Institute of Mental Health (F31MH117827).

Additional support came from a Google Security and Privacy unrestricted gift awarded to JM that provided consulting income for researcher MM. SC, a researcher at Google, is an author on this paper who contributed to study design, analysis, and manuscript preparation. The specific role of this author is articulated in the ‘author contributions’ section. The manuscript was approved by an internal Google review.”

 D2.b. Updated competing interests statement declaring a commercial affiliation 

The support received from Google, the involvement of a researcher employed by Google, and the grant from Samsung do not alter our adherence to PLOS ONE policies on sharing data and materials. 

D3. Grant information in the ‘Funding Information’ and ‘Financial Disclosure’ sections 

The funding from Samsung and Google is now reported in the Funding Information section. There are no grant numbers for these awards.

R2.1 Time 1 versus Time 2 as a point of comparison

R2 raised the concern about the use of Time 1 as the comparison point across years, pointing out that March may have been too early to detect adverse consequences of the pandemic.

We thank R2 for raising this concern. In response, we reran the yearly comparisons based on data from the Time 2 surveys (collected in June of each year). The results were nearly identical suggesting that levels of depression, stress, anxiety, and loneliness were not higher in June of 2020 than in June of 2019. We have added the sentences below to the manuscript and have deleted the two sentences in the limitations that pointed out this issue (Revised Manuscript with Track Changes lines 904-906).

“Time 2 comparisons from 2019 and 2020 similarly indicates no yearly differences in depressive symptoms, anxiety, stress, or loneliness. ” (Manuscript lines 256-257; Revised Manuscript with Track Changes lines 262-263). 

“There were no within-person changes in Time 1 depressive symptoms, stress, or loneliness nor were there any within-person differences in any of these variables, including anxiety, in Time 2 data.” (Manuscript lines 260-262; Revised Manuscript with Track Changes lines 266-268). 

R2.2 Labeling of academic terms. 

R2 pointed out that references to autumn and spring terms may confuse readers located outside the Northern hemisphere. 

To address this potential confusion, we have added dates of the Winter and Spring terms to Table 1, described in R2.3. 

We think the single seasonal label in the text (Spring term) will be less cumbersome for readers than a month and date combination. 

R2.3 Clarification of pandemic timeline and local response 

R2’s suggested that we describe the local and university responses to the pandemic.

In response, we added a timeline to provide context for our findings (Table 1; Manuscript line 133; Revised Manuscript with Track Changes line 136).

Table 1. Timeline of Local Response to the Pandemic 

Date

Event

2/28/2020

Alerts about local community spread of COVID-19.

3/6/2020

University announced shift to online learning for the remainder of Winter term (6 January 2020 - 20 March 2020)

3/13/2020

University announced that Spring term (30 March 2020 - 12 June 2020) would begin online. Statewide ban on large gatherings and closure of K-12 schools. 

3/18/2020

University announced that all of Spring term would be online

3/23/2020

Statewide stay-at-home order issued. 

5/4/2020 - present (March, 2021)

Phased restrictions on in-person contact, based on county public health metrics. Classes remained online for the entire 2020-2021 academic year.

R2.4 Measurement differences in predictor and dependent variables 

R2 asked for clarification on how we reconciled different Likert Scale ratings.

We thank R2 for the opportunity to clarify. For most of the comparisons, we standardized the variables before analyzing the data to account for measurement differences. The coefficients represent standard deviation units rather than the Likert scale ratings, allowing comparison of variables measured on different scales. The models reported in Table 2 are the only ones with unstandardized data. The variables used in these analyses are counts representing the number of active coping behaviors relative to maladaptive. The unstandardized variables allow the reader to see the relative change in distress indicators for each additional adaptive behavior relative to maladaptive ones. These decisions do not change the p-value; they affect only the magnitude of the coefficient. Since we refrain from comparing coefficients across models, we believe that these analytic decisions are sound and provide the reader with the most understandable metrics. 

R2.5. Limitations of self-report

R2 mentioned that self-report data is often called into question and asked about how we handled concerns about self-report data in our analysis. 

We appreciate R2’s concern about self-reported data. We note this as a limitation: "This exploratory study of college student experiences during the pandemic was limited by the self-report nature of assessments, time-based measures as a proxy for pandemic effects, and a college sample … Students' evaluation of their effectiveness in coping was not benchmarked with objective measures such as grade point average." 

(Manuscript lines 860-865; Revised Manuscript with Track Changes lines 889-894)

While coping effectiveness may be less objectively assessed in self-report than by external measures such as grades, internal states such as stress and mood are best assessed by self-report (Stone, 2000). Additionally, a review of the empirical literature on the validity of self-reported health-risk behaviors concludes “self-reported data are accurate when individuals understand the questions and when there is a strong sense of anonymity and little fear of reprisal” (Brener, Billy, & Grady, 2003). Since all survey responses were kept confidential in this study, we have no reason to suspect that participants would be prone to bias their responses. Finally, we tested for the reliability and validity of the self-reported data and include these statistics in the measures section. All measures demonstrate strong psychometric properties, pointing to evidence of minimal response bias. 

One limitation of self-reported internal states concerns retrospective bias, in which recent experiences dominate recall (Tourangeau, 1999). We address this bias through the use of experience sampling methods (ESM) which prompt individuals to describe current and very recent states (Trull & Ebner-Priemer, 2009). Comparisons between ESM data and traditional self-reports find poor to moderate agreement, suggesting inaccuracies in recall of mood and coping styles over long time periods (Stone et al., 1998; Solhan, Trull, Jahng, & Wood, 2009). Our use of ESM may therefore provide a more accurate description of participants’ psychological functioning and coping behaviors than methods relying on retrospective recall. 

References: 

Brener, N. D., Billy, J. O., & Grady, W. R. (2003). Assessment of factors affecting the validity of self-reported health-risk behavior among adolescents: Evidence from the scientific literature. Journal of Adolescent Health, 33(6), 436-457.

Solhan, M. B., Trull, T. J., Jahng, S., & Wood, P. K. (2009). Clinical assessment of affective instability: comparing EMA indices, questionnaire reports, and retrospective recall. Psychological Assessment, 21(3), 425.

Stone, A. A. (2000). The science of self-report: Implications for research and practice. Mahwah, NJ: Lawrence Erlbaum.

Stone, A. A., Schwartz, J. E., Neale, J. M., Shiffman, S., Marco, C. A., Hickcox, M., ... & Cruise, L. J. (1998). A comparison of coping assessed by ecological momentary assessment and retrospective recall. Journal of Personality and Social Psychology, 74(6), 1670.

Tourangeau, R. (1999). Remembering what happened: Memory errors and survey reports. In The science of self-report (pp. 41–60). Psychology Press.

Trull, T. J., & Ebner-Priemer, U. W. (2009). Using experience sampling methods/ecological momentary assessment (ESM/EMA) in clinical assessment and clinical research: introduction to the special section. Psychological Assessment, 21(4), 457–462. https://doi.org/10.1037/a0017653

R2.6 Organization of qualitative themes by challenges and strategies

R2 asked why we chose to organize themes by challenges and strategies. 

To some extent, this structure flowed from the objective of this study. We sought to learn about students’ experiences during the pandemic -- widely recognized as a challenge for students and other populations. In-depth interviews allowed us to characterize those challenges and illustrate how these are contextualized in students’ daily lives. Students’ detailed descriptions of their attempts to learn and communicate in this context illuminated their strategies for academic engagement and psychosocial wellbeing. In the course of analysis, we recognized that most of the content and most of the codes related either to challenges that students faced or how they were coping with those challenges, so it was a natural framework for presenting the themes. We differentiated sets of students’ strategies and described them in detail with the idea that they could inform thinking about interventions. 

R2.8 Inclusion of the interview protocol

The interview protocol has been added to the repository, as requested by R2. 

https://github.com/kskuehn/UWEXP_COVID/blob/main/Interview%20Guide%20for%20PLOS%20ONE%20repository.docx

R2.9 Confusion around whether or not CBT was delivered to participants

R2 requested clarification about CBT offered in this study. 

This study did not involve any treatment or skills training. We do propose broader dissemination of skills training offered in CBT and DBT modalities, based on the findings that active coping was associated with lower depression ratings as well as higher ratings of perceived effectiveness in coping. We have reworded the manuscript for clarity. The original text "To help students develop effective coping skills, the training offered in cognitive behavioral therapy and dialectical behavioral therapies could be made broadly accessible." has been revised to "To help students develop effective coping skills, the skills training that is commonly part of cognitive behavioral and dialectical behavioral therapies could be offered to all students, not just those in treatment for mental health concerns.” 

(Manuscript lines 849-852; Revised Manuscript with Track Changes lines 877-880). 

R2.10 Connection between final discussion points and study findings

R2 raised the issue of whether our final discussion points followed from our findings.

We edited the final paragraph of the discussion to clarify the link with our results. The sentence, “Rather than relying on a one-size-fits-all approach to supporting students who are participating in college remotely, interventions may need to be tailored and prioritized for different individuals, keeping in mind the context for their struggles.” 

is now worded:

 “Rather than relying on a one-size-fits-all approach to supporting students who are participating in college remotely, our findings of heterogeneity underscore the importance of tailoring interventions, keeping in mind the context for their struggles.” 

(Manuscript line 843; Revised Manuscript with Track Changes line 870).

We have confirmed that the remaining discussion points are rooted in our analysis.

R2.11. Production errors and typos

We reviewed the document and corrected errors. Those corrections are marked.

Additional changes

Two minor statistical corrections were made, described below. These do not alter the major findings or the points highlighted in the discussion. 

In the initial draft, when providing the summary statistics for between-person coping as a predictor of distress during the 2020 Spring term, we accidentally reported the results for the Perceived Stress Scale-4 (PSS-4) instead of the Stress Severity Ratings. This error did not affect Figure 3 or the conclusions, which were rightly based on the Stress Severity item. This error has been in this revised submission (Manuscript lines 256-257; Revised Manuscript with Track Changes lines 262-263).

We also noticed that in the initial submission the coping models that predicted academic stress were erroneously reported from a mixed-effects model without the random time effect included. It was determined from fit indices (i.e., AIC and BIC) that this effect should be included and it was included in all of our other coping models. We therefore applied the model with the random effect to the ‘Coping comparisons during Spring 2020 term’ section. With this correction, the between-person problem-focused coping no longer predicted increased academic stress (as it had been reported in the initial submission). The corrected text reports the adjustment (Manuscript lines 343-344; Revised Manuscript with Track Changes lines 354-355).

For clarification, several variables initially only reported in S5 table were added to the text (Manuscript lines 268-272; Revised Manuscript with Track Changes lines 275-279) to describe the small changes in reported distress over the Spring 2020 term. 

Some stylistic changes were made for clarity and one supporting reference was added. Those edits are marked in the manuscript.

One co-author, accidentally omitted in the initial submission, has been added. That change was made in the portal and the title page.

---

## [Decision Letter · Decision Letter 1]

29 Apr 2021

College from home during COVID-19: A mixed methods study of heterogeneous experiences

PONE-D-20-34654R1

Dear Dr. Morris,

We’re pleased to inform you that your manuscript has been judged scientifically suitable for publication and will be formally accepted for publication once it meets all outstanding technical requirements.

Kind regards,

Amanda A. Webster

Academic Editor

PLOS ONE

Additional Editor Comments (optional):

Reviewers' comments:

Reviewer's Responses to Questions

**Comments to the Author**

1. If the authors have adequately addressed your comments raised in a previous round of review and you feel that this manuscript is now acceptable for publication, you may indicate that here to bypass the “Comments to the Author” section, enter your conflict of interest statement in the “Confidential to Editor” section, and submit your "Accept" recommendation.

Reviewer #1: All comments have been addressed

Reviewer #2: All comments have been addressed

2. Is the manuscript technically sound, and do the data support the conclusions?

Reviewer #1: Yes

Reviewer #2: Yes

3. Has the statistical analysis been performed appropriately and rigorously? 

Reviewer #1: Yes

Reviewer #2: Yes

4. Have the authors made all data underlying the findings in their manuscript fully available?

Reviewer #1: Yes

Reviewer #2: Yes

5. Is the manuscript presented in an intelligible fashion and written in standard English?

Reviewer #1: Yes

Reviewer #2: Yes

6. Review Comments to the Author

Reviewer #1: Thank you very much for the detailed response and the corrections. All issues have been addressed and the structure is clearer now. I recommend this paper for publication.

Reviewer #2: I appreciate that the authors have thoroughly addressed questions and comments from the earlier review.

I noted a couple of typos: Lines 268, 280, p. 12: words missing before naming tables (S5 table, S6-S8 tables)

7. PLOS authors have the option to publish the peer review history of their article (what does this mean?). If published, this will include your full peer review and any attached files.

Reviewer #1: **Yes: **Dimitrios Vlachopoulos

Reviewer #2: No

---

## [Editor Report · Acceptance letter]

18 Jun 2021

PONE-D-20-34654R1 

College from home during COVID-19: A mixed-methods study of heterogeneous experiences 

Dear Dr. Morris:

I'm pleased to inform you that your manuscript has been deemed suitable for publication in PLOS ONE. Congratulations! Your manuscript is now with our production department. 

Kind regards, 

on behalf of

Dr. Amanda A. Webster 

Academic Editor

PLOS ONE